# Nicotinic Receptor Subunits Atlas in the Adult Human Lung

**DOI:** 10.3390/ijms21207446

**Published:** 2020-10-09

**Authors:** Zania Diabasana, Jeanne-Marie Perotin, Randa Belgacemi, Julien Ancel, Pauline Mulette, Gonzague Delepine, Philippe Gosset, Uwe Maskos, Myriam Polette, Gaëtan Deslée, Valérian Dormoy

**Affiliations:** 1Inserm UMR-S1250, P3Cell, University of Reims Champagne-Ardenne, SFR CAP-SANTE, 51092 Reims, France; zania.diabasana@inserm.fr (Z.D.); jmperotin-collard@chu-reims.fr (J.-M.P.); randa.belgacemi@univ-reims.fr (R.B.); jancel@chu-reims.fr (J.A.); pmulette@chu-reims.fr (P.M.); gdelepine@chu-reims.fr (G.D.); myriam.polette@univ-reims.fr (M.P.); gdeslee@chu-reims.fr (G.D.); 2Department of Respiratory Diseases, Centre Hospitalier Universitaire de Reims, Hôpital Maison Blanche, 51092 Reims, France; 3Department of Thoracic Surgery, Centre Hospitalier Universitaire de Reims, Hôpital Maison Blanche, 51092 Reims, France; 4CNRS UMR9017, Inserm U1019, University of Lille, Centre Hospitalier Régional Universitaire de Lille, Institut Pasteur, CIIL—Center for Infection and Immunity of Lille, 59000 Lille, France; philippe.gosset@pasteur-lille.fr; 5Integrative Neurobiology of Cholinergic Systems, Institut Pasteur, CNRS UMR 3571, 75015 Paris, France; uwe.maskos@pasteur.fr; 6Department of Biopathology, Centre Hospitalier Universitaire de Reims, Hôpital Maison Blanche, 51092 Reims, France

**Keywords:** nicotinic receptors, airway epithelial cells, lung

## Abstract

Nicotinic acetylcholine receptors (nAChRs) are pentameric ligand-gated ion channels responsible for rapid neural and neuromuscular signal transmission. Although it is well documented that 16 subunits are encoded by the human genome, their presence in airway epithelial cells (AECs) remains poorly understood, and contribution to pathology is mainly discussed in the context of cancer. We analysed nAChR subunit expression in the human lungs of smokers and non-smokers using transcriptomic data for whole-lung tissues, isolated large AECs, and isolated small AECs. We identified differential expressions of nAChRs in terms of detection and repartition in the three modalities. Smoking-associated alterations were also unveiled. Then, we identified an nAChR transcriptomic print at the single-cell level. Finally, we reported the localizations of detectable nAChRs in bronchi and large bronchioles. Thus, we compiled the first complete atlas of pulmonary nAChR subunits to open new avenues to further unravel the involvement of these receptors in lung homeostasis and respiratory diseases.

## 1. Introduction

Nicotinic acetylcholine receptors (nAChRs) are ligand-gated (cation-permeable) proteins expressed in the brain and non-neuronal cells, including lung airway epithelial cells (AECs), macrophages, neutrophils, and muscle cells [1]. These receptors are composed of five subunits organized as homo- or hetero-pentamers forming a channel permeable to monovalent and divalent cations (predominantly Na^+^, K^+^, and Ca^2+^) [1,2]. There are 16 mammalian subunits, namely α1–α7, α9–α10, β1–β4, δ, ε, and γ (the corresponding gene names are, respectively, CHRN (cholinergic receptors nicotinic subunits) A1–A7, A9–A10, B1–B4, D, E, and G) [2]. Each subunit shares a common structure comprising a large amino-terminal segment (about 200 residues), four transmembrane domains TM1, TM2, TM3, and TM4 (less than 30 residues each), a large cytoplasmic loop (90 to 270 residues) localized between TM3 and TM4, and a variable carboxyl tail (10 to 30 residues) [2,3,4]. Muscle-type nAChRs are generally assembled from 2(α1)/β1/δ/γ or 2(α1)/β1/δ/ε subtypes depending on muscle innervation [4,5,6,7]. Neuronal nAChRs are assembled from α2–α7, α9–α10, and β2–β4 [8,9,10]. In the brain, homomeric α7 and heteromeric α4/β2 subtypes are abundantly detected and are known to play an important role in memory and learning due to their predominance in the hippocampal and cortical neurons [11,12]. Other subtypes such as homomeric α9, heteromeric α2/α6, α7/β2, and α9/α10 have also been detected to a lesser extent [1,2,3,12,13,14,15,16,17,18]. The diversity of nAChRs confers differential affinities to the ligands affecting several parameters, including the channel opening and closing duration, the modulation of ionic conductance, and cationic selectivity [1,2,12,14].

From a functional perspective, acetylcholine binding to nAChR at the extracellular interface between two subunits leads to an allosteric conformational change permitting the channel opening, followed by ion fluxes across the plasma membrane that participate in cell survival, differentiation, and proliferation [3,9,19,20]. Nicotine, one of the components of cigarette smoke, acts as an agonist implicated in the inhibition of apoptosis and oxidative stress responses, ultimately leading to lung impairments due to long-term exposure [4,21,22]. Although nicotinic receptors are ionotropic complexes, they may display metabotropic signalling properties via their association with trimeric GTP-binding proteins to regulate downstream pathways and cytokine expression [23,24].

Previous studies have established that multiple single-nucleotide nAChR polymorphisms are associated with risks of lung cancer and chronic obstructive pulmonary disease (COPD), highlighting their potential implication in respiratory diseases [19,25,26]. In addition, it has been hypothesized that the nAChRs may play a role in coronavirus disease 2019 (COVID-19) and might represent a therapeutic target, particularly regarding their potential contribution in the regulation of angiotensin-converting enzyme-2 (ACE-2), the main receptor for severe acute respiratory syndrome coronavirus (SARS-CoV-2) [27,28,29]. Altogether, this underlines the requirement of deciphering the atlas of pulmonary nAChR subunits.

Indeed, if the general expressions of muscle and neuronal nAChRs are well known, little information is available regarding their expression in the lung and particularly in different AECs [30]. Therefore, we conducted a transcriptomic and proteomic analysis of the localization and expression of all human nAChR subunits in the adult lung.

## 2. Results

### 2.1. Smoking-Associated Pulmonary nAChR Subunit Transcript Expressions

Considering the whole lung contains all types of tissues, including epithelia, muscle, connective, and nervous tissues (Figure 1a), the 16 nAChRs were detected among non-smoker subjects except for CHRNA7, which was consistent in both datasets containing non-smokers (Figure 1b and Appendix A). CHRNB1/E were very highly expressed; CHRNA6/A9/A10/B3/D were highly expressed (see Section 4). There was a significant increase in CHRNA1/A2/A7/B3/B4 transcript levels in smokers compared to non-smokers. Interestingly, CHRNA7 was only detected in smokers. On the contrary, CHRNA3/A4/A9/B2/D/G transcript levels were significantly decreased in smokers. The global repartition in non-smokers and smokers favoured CHRNA10/B1/E, representing almost half of all nAChR subunits expressed in lung tissues (Figure 1c and Appendix A). The differential repartitions of nAChRs between non-smokers and smokers matched their differential expressions.

In large AEC (LAEC) (Figure 2a) CHRNA1/A2/A4/B1/B3/B4/D were not detected in non-smokers (Figure 2b and Appendix A). CHRNA5 was very highly expressed; CHRNA7/A10 were highly expressed. Interestingly, CHRNB1/B4 were only detected in smokers. There was a significant decrease in CHRNA5/A10 transcript levels in smokers when compared to non-smokers. The global repartition in non-smokers and smokers favoured CHRNA5/A7/A9/A10, representing more than 75% of all nAChRs expressed in LAEC (Figure 2c and Appendix A). There were no significant differences in terms of nAChR repartitions between non-smokers and smokers.

In small AEC (SAEC) (Figure 3a), the 16 nAChRs were detected among non-smokers with moderate or low expressions (Figure 3b and Appendix A). There was a significant increase in CHRNA5/A7/B2/B3 transcript levels in smokers compared to non-smokers. The global repartition in non-smokers and smokers favoured CHRNA7/A9/A10/B2, representing half of the nAChRs expressed in SAEC (Figure 3c and Appendix A). There was a significant increase in CHRNA7/B2/B3 and a significant decrease in CHRNA2/A9 in the repartition of nAChRs in smokers compared to non-smokers.

### 2.2. Differential Pulmonary nAChRs Transcript Expressions at the Single-Cell Scale

At the level of single-cell transcriptomes (Figure 4), CHRNA5/A7/A9/A10/B1/E were highly expressed in most of the AEC populations including alveolar, basal, goblet, multiciliated, and club cells. CHRNA1/A2/A4/A6/B2/B3/D/G showed low to no expression in AEC. Interestingly, functional AEC cell populations were distinguished with their nAChR signatures: pneumocytes expressed CHRNA5/A10/B1; basal cells expressed CHRNA5/A7/A10/B1/E; goblet cells expressed CHRNA7/A10/B1/E; multiciliated cells expressed CHRNA9/10/B1/E; club cells expressed CHRNA7/A10/B1; ionocytes expressed CHRNA3/B4/E.

Considering non-epithelial cells, CHRNA1/A3/A5/A10/B1/E were highly expressed in specific populations of immune cells including macrophages, B cells, dendritic cells, and mast cells. CHRNB1 expression was specific to lymphatic cells. CHRNA5/B1/E were highly expressed in fibroblasts; CHRNA3/A10/B1 in smooth muscle cells; CHRNA5/B1/E in endothelial cells and macrophages. B cells and dendritic cells mainly expressed CHRNB1/E.

### 2.3. Identification of nAChR Subunits in Bronchial and Large Bronchiolar Epithelia

To investigate nAChR subunit localization in the lung, we selected commercially available validated primary antibodies displaying the antigenic sequences demonstrating the lowest percentage of identity with regard to cross-reactivity (Appendix A). We focussed here on bronchi and large bronchioles as well as smooth muscle and blood vessels on formalin-fixed paraffin-embedded (FFPE) tissues (Figure 5a,b). Subunits α1/α2/α4/β3/γ were not detected. Subunit α3 seemed restricted to the apical side of differentiated cells. Surprisingly, α5 was systematically found in AEC nuclei and the apical side of differentiated AECs, while its pattern was consistent with membrane-bound receptors on smooth muscle cells. Subunits α6 and α9 presented similar staining in differentiated AECs, such as α7/α10/β1/β2/δ/ε, which in addition were found in non-differentiated AECs. Finally, β4 appeared in multiciliated cells only. When available, our observations were generally concordant with the data from the Human Protein Atlas (Figure 5b and Appendix A).

## 3. Discussion

This is the first study showing transcript levels and localizations of all nAChR subunits in the human adult lung. Interestingly, we identified distinct variations in terms of nAChR transcript levels between whole-lung tissues, LAEC, and SAEC, as well as important changes between non-smokers and smokers. Since whole-lung transcriptomics encompasses all pulmonary tissues, isolated cell studies represent the ideal strategy to unveil nAChR functions in airways. It has been successfully implemented in the context of AEC differentiation analysis, asthma, and idiopathic pulmonary fibrosis [31,32,33,34,35]. If they summarize the transcriptomic profile of the organ, whole-lung microarray data require tissue or single-cell analyses to distinguish the contribution of each cellular population to the specific gene expression. Otherwise, it would be admitted that a gene is ubiquitously expressed in the lung, while it is only found in one histological tissue. As such, our comparative analysis pointed mainly towards CHRNA5/A7/A10/B2/B3 to tackle the association of nAChR expression and smoking. Furthermore, the impact of smoking could be tied into the associated risks of respiratory diseases, provided complete clinical data are available.

We included in the analysis of nAChR expression levels 298 subjects in three distinct modalities (whole-lung tissues, LAEC, and SAEC) and performed a preliminary identification of single-cell transcriptomic signature. Our immunostaining analyses provided important data regarding the subcellular localization of nAChR subunits in bronchi and large bronchioles. Microscopic observations and transcriptomic analysis were generally concordant. Because of their modalities of association at the cell membrane and their high degree of amino acids identity, nAChR immunostainings were generally sparse, rarely concordant, and performed on murine tissues in the literature [30,36,37,38]. A careful validation method including heterologous cells overexpressing the different human nAChR subtypes is required to further validate all subunit nAChR antibodies in the human adult lung [39]. Nonetheless, since we selected all our antibodies based on thorough sequence alignments of the antigenic sequences, we provided here a complete description of all nAChRs in bronchi and large bronchioles. Only individual subunits were detected and not the receptors, which are assembled of five subunits; in vitro experimental studies will be required to confirm the presence of various pentamers at the cell surface of the lung tissues. Other caveats complicating the identification of nAChRs include their dynamic of assembly/recycling at the cell surface [7,40] and their differential requirement according to the cellular context (quiescence, proliferation, oxidative stress, etc.) [1].

Additional studies on larger cohorts are needed to complement and refine our analysis. Deciphering the cellular and molecular impact of the observed differences in transcript expressions in the context of smoking will be essential to understanding nAChR-associated pathogenesis. It will be particularly insightful for at least three lung diseases where smoking may partly impact homeostasis: lung cancers, COPD, and COVID-19. (i) nAChR single-nucleotide polymorphisms (SNPs) were associated with lung cancer cells [41,42], and nAChRs were shown to be involved in cancer cell proliferation and survival [43,44,45,46]. Interestingly, several subunits (including α5-7-10/β2-3) were identified in cancer cell lines, and selective nAChR inhibitors induced anti-tumour effects [47,48]. In addition, acetylcholine-signalling proteins were involved in the progression of lung cancer [49]. Altogether, understanding the repartition and the possible assembly of nAChRs at the cancer cell surface may pave the way towards the design of effective anti-cancer drugs. (ii) nAChR SNPs were also associated with nicotine dependence and COPD [50,51,52,53,54]. CHRNA3/A5/B4 polymorphisms were heavily discussed in the dissection of the genetic origins of COPD, but no functional studies have been published so far [55]. In addition, α7 and its ligands received particular attention as potential inflammatory players in COPD patients [54,56]. Exploring the involvement of nAChRs in COPD pathogenesis and progression in light of their differential distribution in lung cell populations may help improve health care for this pathology lacking treatments. (iii) Nicotine, an exogenous ligand of nAChRs and, more generally, smoking, have been shown in vitro and in vivo to modulate the expression of hACE2, the main receptor of the SARS-CoV-2 spike S protein [57,58,59,60]. In light of the differential subunit expressions, it will be of interest to analyze the localizations of hACE2 and nAChRs in COVID-19 tissues.

We provided the first atlas of nAChR subunits in the lung and invited cartographers to complete the map in order to provide a fundamental understanding of these crucial actors of homeostasis that may contribute to chronic and acute respiratory diseases. The identification of each potential subunit that may assemble functional channels at the cell surface is a requisite for the optimal design of efficient pharmacological modulations of nAChRs in the context of the pharmacology of the respiratory system.

## 4. Materials and Methods

### 4.1. Human Subjects

Patients scheduled for lung resection for cancer (University Hospital of Reims, France) were prospectively recruited (*n* = 10) following standards established and approved by the institutional review board of the University Hospital of Reims, France (IRB Reims-CHU, date of approval: 12 June 2011). In addition, 10 patients who underwent a routine large airway fiberoptic bronchoscopy with bronchial brushings under local anaesthesia according to international guidelines were also recruited (5 non-smokers, 5 smokers) [61]. Informed consent was obtained from all the patients. Subjects were recruited from the Department of Pulmonary Medicine at the University Hospital of Reims (France) and included in the cohort for Research and Innovation in Chronic Inflammatory Respiratory Diseases (RINNOPARI, NCT02924818). The study was approved by the ethics committee for the protection of human beings involved in biomedical research (CCP Dijon EST I, N°2016-A00242-49, date of approval: 31 May 2016) and was conducted in accordance with the ethical guideline of the Declaration of Helsinki. Patients with chronic obstructive pulmonary disease, asthma, cystic fibrosis, bronchiectasis, or pulmonary fibrosis were excluded. At inclusion, age, sex, smoking history, and pulmonary function test results were recorded to exclude patients with an alteration of lung functions. Ex-smokers were considered for a withdrawal longer than 5 years.

### 4.2. Sample Processing

Fresh airway epithelial cells (AECs) obtained from bronchial brushings (right lower lobe, 5th to 8th divisions) were suspended for 30 min in Roswell Park Memorial Institute Medium (RPMI) (1% penicillin/streptomycin + 10% Bovine Serum Albumin (BSA)) before centrifugation (13,500*g* ×2 times). The cell pellet was dissociated in 1 mL of Trypsin Versene (Lonza), centrifuged (13,500*g* ×2 times), and kept at −20 °C until further steps.

### 4.3. RT-qPCR Analyses

Total RNA from AEC bronchial brushings was isolated by a High Pure RNA isolation kit (Roche Diagnostics), and 250 ng was reverse-transcribed into cDNA by a Transcriptor First Stand cDNA Synthesis kit (Roche Diagnostics, Meylan, France). Quantitative PCR reactions were performed with a Fast Start Universal Probe Master kit and UPL-probe system in a LightCycler 480 Instrument (Roche Diagnostics) as recommended by the manufacturer. Primers listed in Appendix A were designed via the Universal Probe Library Assay Design Center (Roche, Manheim, Germany). Results for all expression data regarding transcripts were normalized to the expression of the house-keeping gene GAPDH amplified with the following primers: forward 5′-ACCAGGTGGTCTCCTCTGAC-3′, reverse 5′-TGCTGTAGCCAAATTCGTTG-3′. We verified that GAPDH transcript detection levels were highly similar between non-smokers and smokers to validate the housekeeping gene (average Ct = 25.54 ± 0.17 in non-smokers vs. 25.35 ± 0.34 in smokers; *p* = 0.64). Relative gene expression was assessed by the ΔΔCt method [62] and expressed as 2^−ΔΔCt^. To compare data generated via PCR with RNAseq analysis, we transformed the transcript expressions to a percentage scale considering the highest and lowest values per subunit for the detection, or across all the subunits for the repartition.

### 4.4. Immunofluorescent Staining and Analyses

Immunohistochemistry was performed on formalin-fixed paraffin-embedded (FFPE) lung tissues distant from the tumour as previously described [63]. Only patients having no respiratory diseases were included (smokers and ex-smokers). Five micrometer sections were processed for hematoxylin and eosin staining and observed on a microscope (×20) to confirm the presence of bronchi and large bronchioles (pseudostratified epithelia). The bronchial epithelium was analysed on the entire slide including 2 to 7 units per patient. FFPE lung tissue section slides were deparaffinised and blocked with 10% BSA in phosphate-buffered saline (PBS) for 30 min at room temperature. Tissue sections were then incubated with the primary antibodies as listed in Appendix A for one night at 4 °C in 3% BSA in PBS. After the PBS wash, a second primary antibody was used to highlight non-differentiated cells on epithelia for 2h at room temperature: p63 (AF1916, 1:200, R&D Systems, Noyal Châtillon sur Seiche, France) or pan-cytokeratin (CK, 1:1000, E-AB-33599, Elabscience, Clinisciences, Nanterre, France). Sections were washed with PBS and incubated with the appropriate secondary antibodies in 3% BSA in PBS for 30 min at room temperature: Alexa Fluor^®^ (Invitrogen, Fisherscientific, Illkirch, France) donkey anti-rabbit IgG 594 (A21207), donkey anti-goat IgG 488 (A11055), goat anti-mouse IgG 594 (A11005), and goat anti-rabbit IgG 488 (A11008). DNA was stained with DAPI during incubation with the secondary antibodies. Micrographs were acquired on a Zeiss AxioImageur (20× Ph) with ZEN software (8.1, 2012) and processed with ImageJ (National Institutes of Health) for analysis. For each patient, five random fields per section containing bronchi and large bronchioles were taken to evaluate the localization of nicotinic receptors on epithelial and stromal cells. We selected the most suitable primary antibodies directed against each subunit, considering external validations, identity, and staining optimization, to highlight the localization of nAChRs on bronchi and large bronchioles.

### 4.5. Transcriptome Profiling Microarray Analysis

Gene expressions of non-smoking and smoking subjects with no chronic respiratory diseases were collected from datasets available online (GEO database; http://www.ncbi.nlm.nih.gov/geo) including whole-lung tissue samples in 153 subjects (42 non-smokers, 111 smokers; GSE103174, 76925, and 47460) or small airway bronchoscopic samples (10th to 14th divisions) in 135 subjects (63 non-smokers, 72 smokers; GSE11784).

In order to compare transcriptomic data extracted from various datasets or PCR reactions, we formatted the absolute values to a percentage scale. Concerning the detection of genes, we first identified for each gene the highest and lowest expression values in both non-smokers and smokers to set the maximal value at 100%. After proportionally expressing each of the single expression values for all the subunits, the average was calculated and plotted on a graph. To discuss the relative level of expressions, we arbitrarily categorized 4 groups: (1) very high expressions, the average percentage of expression is over 75% of the maximum; (2) high expressions, the average percentage of expression is between 50 and 75% of the maximum; (3) moderate expressions, the average percentage of expression is between 25 and 50% of the maximum; (4) and low expression, the average percentage of expression is below 25% of the maximum. Concerning the repartition, the total expressions of absolute values for all nAChR were summed for each patient of the considered dataset to express the proportion of each subunit. The comparative average percentage of expression of each subunit for all patients was plotted in a pie chart.

### 4.6. Single-Cell Sequencing

The published dataset can be found at lungcellatlas.org and https://www.covid19cellatlas.org. We retained cell clustering based on the original studies and considered only lung samples (brushing and parenchyma from resected tissues) from subjects with no respiratory disease [33]. An Illumina Hiseq 4000 per 10× Genomics chip position was used (*n* = 6; 2000–5000 cells/sample). Additional sequencing was performed to obtain coverage, or at least mean coverage, of 100,000 reads per cell.

### 4.7. Statistics

The data are expressed as mean values and percentages. Differences between the two groups (non-smokers and smokers) for each gene were determined using the Student *t* test. A *p*-value < 0.05 was considered significant; *, *p* < 0.05; **, *p* < 0.01; ***, *p* < 0.001.

## Figures and Tables

**Figure 1 ijms-21-07446-f001:**
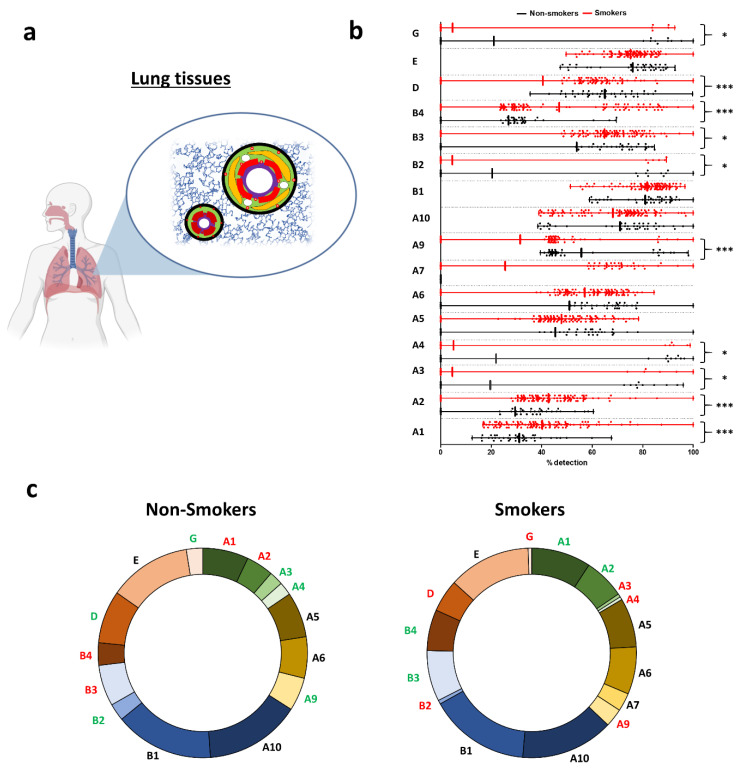
Evaluation of nicotinic acetylcholine receptor (nAChR) transcript levels in human lung tissues. (**a**) Illustration depicting the origin of the samples. Whole-lung tissues contained all the tissues present in the lung either in parenchyma or in/around bronchi and bronchioles. (**b**) Histogram showing the detection of nAChRs in non-smokers (black) and smokers (red). * *p* < 0.05; *** *p* < 0.001 non-smokers (*n* = 42) vs. smokers (*n* = 111). (**c**) Pie charts showing the repartition of nAChRs in non-smokers (left) and smokers (right). Coloured subunits indicate upregulation (green) and downregulation (red) in both groups when statistically significant.

**Figure 2 ijms-21-07446-f002:**
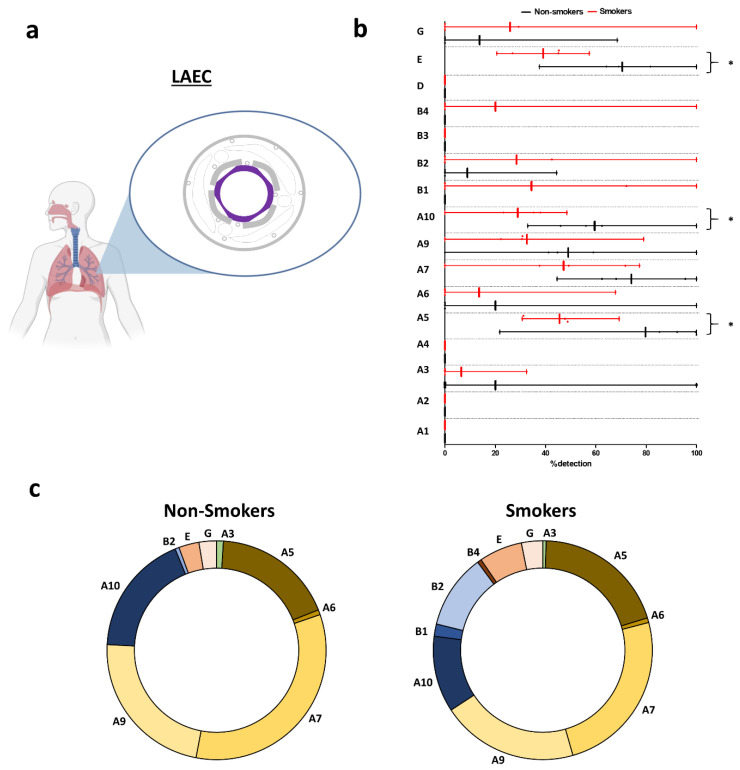
Evaluation of nAChR transcript levels in human large airway epithelial cells (LAECs). (**a**) Illustration depicting the origin of the samples. Isolated AECs were collected from bronchi. Large airway epithelial cells (LAECs) are depicted in purple. (**b**) Histogram showing the detection of nAChRs in non-smokers (black) and smokers (red). * *p* < 0.05 non-smokers (*n* = 5) vs. smokers (*n* = 5). (**c**) Pie charts showing the repartition of nAChRs in non-smokers (left) and smokers (right).

**Figure 3 ijms-21-07446-f003:**
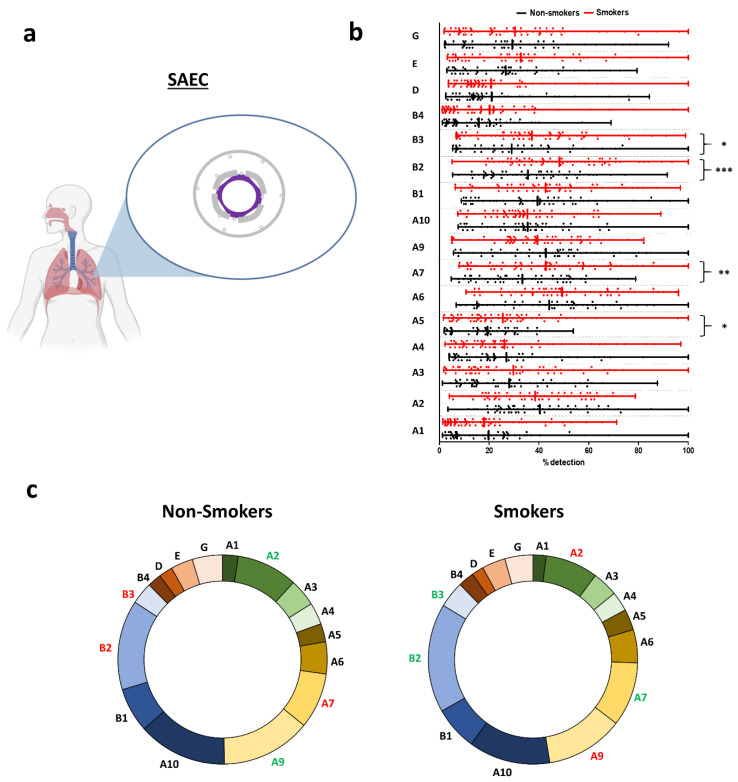
Evaluation of nAChR transcript levels in human small airway epithelial cells (SAECs). (**a**) Illustration depicting the origin of the samples. Isolated AECs were collected from bronchioles. Small airway epithelial cells (SAECs) are depicted in purple. (**b**) Histograms showing the detection of nAChRs in non-smokers (black) and smokers (red). * *p* < 0.05; ** *p* < 0.01; *** *p* < 0.001 non-smokers (*n* = 63) vs. smokers (*n* = 72). (**c**) Pie charts showing the repartition of nAChRs in non-smokers (left) and smokers (right). Coloured subunits indicate upregulation (green) and downregulation (red) in both groups when statistically significant.

**Figure 4 ijms-21-07446-f004:**
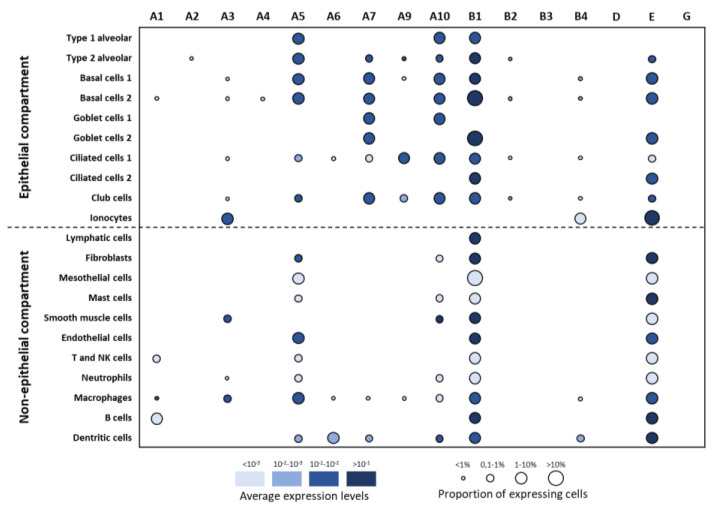
Evaluation of nAChR expressions in lung single-cell populations. Dot plots of nAChR expressions in the epithelial and non-epithelial compartments. The identities of cell populations are shown on the y-axis, and the subunits on the x-axis. The colour intensity represents the average expression level, and the size of the dots represents the proportion of the expressing cells in each population. Raw expression values were normalized, log-transformed, and summarized by published cell clustering.

**Figure 5 ijms-21-07446-f005:**
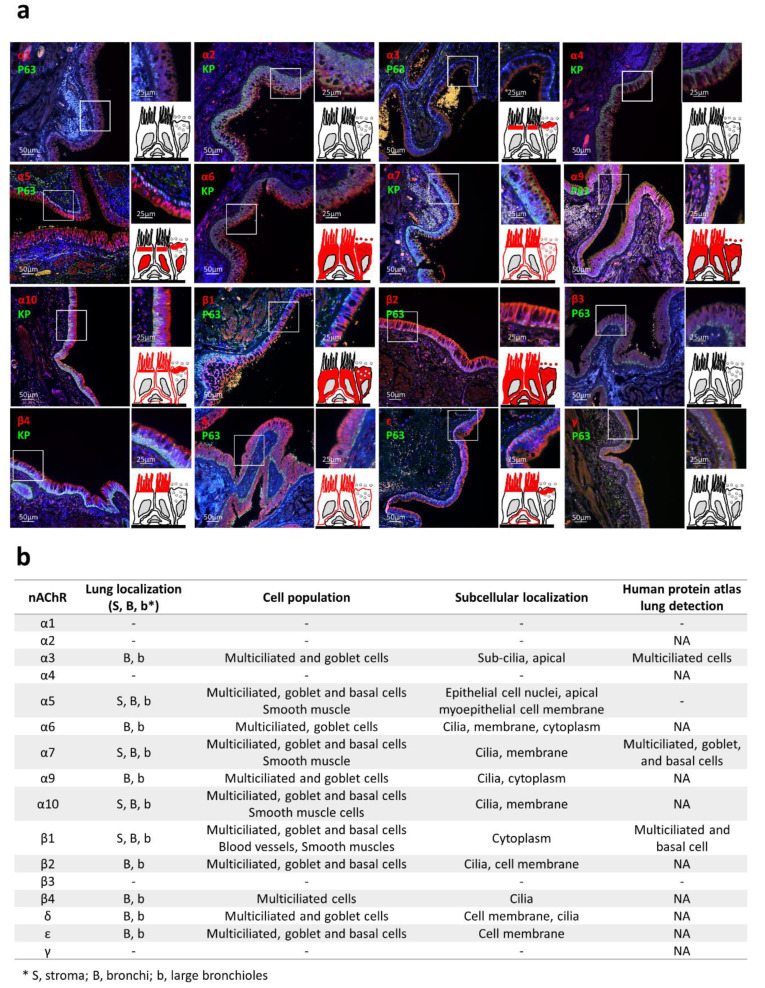
nAChR localizations in human respiratory epithelia. (**a**) Representative micrographs showing the bronchial epithelia on formalin-fixed paraffin-embedded (FFPE) lung tissues stained for the nAChRs (all red), non-differentiated cells (p63 or pan-cytokeratin, green), and cell nuclei (DAPI, blue). Magnification corresponding to the selected area is shown. Drawings depict the localization of each nAChR subunit (in red). (**b**) Table summarizing nAChR subunit cellular and sub-cellular localization and the available microscopic data from the Human Protein Atlas (https://www.proteinatlas.org/). NA, not available; -, no detection.

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
