# Peer review of "Nicotinic Receptor Subunits Atlas in the Adult Human Lung"

_ijms, 2020, doi:10.3390/ijms21207446_

Round 1
Reviewer 1 Report
This was generally a well constructed and implemented study of nicotinic subunits in the human lung. The analysis of the transcriptome as well as the proteome is a strong point of the study. The findings are clear and highly relevant across the nicotinc field, particularly with respect to nictoinic receptors to smoking induced lung cancers. I found it particularly surprising that alpha7 receptors were not detected in non-smokers. I found no issues with the methodology or controls used in the study.
While I found the data extremely well presented, the discussion of the data in terms of human disease was somewhat lacking. There is minimal discussion of the importance of the specific discoveries delineated. In particular, how does the repartitioning of the different subunits correlate with data regarding the importance of these subunits in disease progression? For example, which receptor subtypes are particularly associated with ACE2 regulation or with specific forms of lung cancer. The inclusion of COVID-19, while timely, seemed a bit irrelevant to the findings as no specific COVID-19 tissues were included in the study and no conclusions regarding nicotinic receptors in COVID-19 are presented.
Author Response
This was generally a well constructed and implemented study of nicotinic subunits in the human lung. The analysis of the transcriptome as well as the proteome is a strong point of the study. The findings are clear and highly relevant across the nicotinc field, particularly with respect to nictoinic receptors to smoking induced lung cancers. I found it particularly surprising that alpha7 receptors were not detected in non-smokers. I found no issues with the methodology or controls used in the study.
> We thank the Reviewer for his interest in our findings.
As an additional comment we would like to point out that concerning a7 subunit, the absence in non-smokers is only in lung tissues, they were found in LAEC and SAEC, suggesting that lung tissues are enriched with connective tissues reducing the available epithelial material for analysis. This is corroborated by the results obtained by the human cell atlas consortium, in which the percentage of AEC is low in bronchial tissue or in lung parenchyma. In addition, the potential detection of low expressed transcripts may be restrained by their differential expressions along the airways from the main bronchi until the alveoli.
While I found the data extremely well presented, the discussion of the data in terms of human disease was somewhat lacking. There is minimal discussion of the importance of the specific discoveries delineated. In particular, how does the repartitioning of the different subunits correlate with data regarding the importance of these subunits in disease progression? For example, which receptor subtypes are particularly associated with ACE2 regulation or with specific forms of lung cancer. The inclusion of COVID-19, while timely, seemed a bit irrelevant to the findings as no specific COVID-19 tissues were included in the study and no conclusions regarding nicotinic receptors in COVID-19 are presented.
> We agree with the Reviewer and improve this aspect with the following modifications:
(i) We discussed the implication of our findings with additional comments and references lines 198-208.
(ii) Since we do not present data concerning COVID-19 samples, we removed some irrelevant comments on the actual pandemic with a suggestion that may prove useful to understand the mechanisms of virus entry in this context lines 210-212.
Reviewer 2 Report
In this paper, the authors show the expression and localization of all nAChR subunits in the human adult lung. They identified distinct variations in terms of nAChR transcript levels between whole lung tissues, small and large airway epithelial cells and between smokers and non-smokers. Furthermore by immunostaining analysis the subcellular localization of various nAChR subunits was performed in bronchi and large bronchioles.
All the experiments presented in the manuscript have been undertaken carefully and constitute an important step forward the better understanding of the role of nicotinic receptors in lung function and homeostasis and their potential involvement in different respiratory diseases.
My only comment concerns the fact that in this study only individual subunits were analyzed without any information on the composition and stochiometry of the functional pentameric receptor at the cell surface.
How the authors explain for example in the whole lung of smokers the increase of alpha1 transcript while delta and gamma decrease ? It seems hardly compatible with variation in functional muscle-type receptor expression.
Furthermore, surprinsingly some subunits absent (a7) or present at very low levels (a5, a9) in the whole lung of non-smokers are highly expressed or even the most abundant in isolated LAEC and SAEC. How interpret these observations ?
Minor point :
In the Table S8, the size of the antigenic sequence of a7 subunit does not match with the position 52-259
Author Response
In this paper, the authors show the expression and localization of all nAChR subunits in the human adult lung. They identified distinct variations in terms of nAChR transcript levels between whole lung tissues, small and large airway epithelial cells and between smokers and non-smokers. Furthermore by immunostaining analysis the subcellular localization of various nAChR subunits was performed in bronchi and large bronchioles.
All the experiments presented in the manuscript have been undertaken carefully and constitute an important step forward the better understanding of the role of nicotinic receptors in lung function and homeostasis and their potential involvement in different respiratory diseases.
> We thank the Reviewer for his interest in our findings.
My only comment concerns the fact that in this study only individual subunits were analyzed without any information on the composition and stochiometry of the functional pentameric receptor at the cell surface.
> We agree but this is a limitation that we discussed lines 187-191. We aim to take this extra step in the next few years.
How the authors explain for example in the whole lung of smokers the increase of alpha1 transcript while delta and gamma decrease ? It seems hardly compatible with variation in functional muscle-type receptor expression.
Furthermore, surprinsingly some subunits absent (a7) or present at very low levels (a5, a9) in the whole lung of non-smokers are highly expressed or even the most abundant in isolated LAEC and SAEC. How interpret these observations ?
> Whole lung data are interesting, but as we mentioned in the manuscript, the main issue is the diversity of tissues and cell types. In particular epithelial cells will roughly represent at most the half of the cells taken into consideration for the analysis (connective tissues will be too preponderant). Thus, particularly regarding to functional muscle-type receptor expression, whole lung tissues would not be an adequate indicator. The same applies for some subunits that may appear absent or present at low levels in whole lung tissues while they will be detected in isolated epithelial cells. That is the reason why we suggested to systematically resort to single cell sequencing in order to address the various lung populations.
Minor point :
In the Table S8, the size of the antigenic sequence of a7 subunit does not match with the position 52-259
We thank Reviewer for spotting this error and we verified all of the sequences.
The correct position for a7 was 58-149, it has been corrected in Table S8. We added minor corrections on B1, B4 and D.
Reviewer 3 Report
The study shows different patterns of nAChR subunits expression in different human lung epithelium cells, evaluates nAChR expression in airway epithelial cells, and shows the localization of different nAChR subunits in human respiratory epithelia. The study can provide new insights into the regulation of airway epithelial homeostasis in health and disease, which is especially actual during the COVID-19 pandemic.
The research design is appropriate, conclusions logically follow from either in silico analysis or the experimental results. The study is well-written and in general is easy to read.
It also should be noted that an article was significantly improved in comparison with the version posted on the biorxiv.
However, there are some minor points which should be addressed before article acceptance:
1. The Figs 1,2 and 3 should be done more friendly to the reader: “smokers” and “non-smokers” should be decrypted as the legend in “b” panels, Median % of detection should be shown more clearly, and statistical significance of % of detection from table S1/S3/S5 should be shown directly on “b” panel of figures. You also should consider to emphasize up- or down- regulation of nAChR subunits expression on “c” panels by highlighting subunits font as green or red according to table S2/S4/S6 data.
2. You should decipher the abbreviation CHRNA in results (better in the introduction)
3. Please, provide criteria for “very high” or “high” nAChR expression on lines 75 and 76 or indicate that it is mentioned in 4.5. Methods section.
4. Some issues about immunohistochemistry:
a) Could you provide isotype control/secondary Ab staining of a specimen as a supplementary figure?
b) Could you justify usage of p63 or pan-cytokeratin as a stem cell marker for co-staining of different nAChRs?
c) Ab specificity analysis is impressive but table S8 data leads to the question: how could specific stain be guarantied in pairs: α3-nAChR and α6-nAChR; α9-nAChR and α10-nAChR; β2-nAChR and β4-nAChR?
d) Please, provide scales for all ICH pics (including S2, if possible).
5. Methods section:
a) Please, provide G instead of rpm in 4.2. section.
b) For qPCR primers provide references (if possible) and amplicon length.
c) Provide dilutions for p63 and cytokeratin Abs. Add cat# and dilutions for secondary Abs
Author Response
The study shows different patterns of nAChR subunits expression in different human lung epithelium cells, evaluates nAChR expression in airway epithelial cells, and shows the localization of different nAChR subunits in human respiratory epithelia. The study can provide new insights into the regulation of airway epithelial homeostasis in health and disease, which is especially actual during the COVID-19 pandemic.
The research design is appropriate, conclusions logically follow from either in silico analysis or the experimental results. The study is well-written and in general is easy to read.
It also should be noted that an article was significantly improved in comparison with the version posted on the biorxiv.
> We thank the Reviewer for his interest in our findings.
However, there are some minor points which should be addressed before article acceptance:
- The Figs 1,2 and 3 should be done more friendly to the reader: “smokers” and “non-smokers” should be decrypted as the legend in “b” panels, Median % of detection should be shown more clearly, and statistical significance of % of detection from table S1/S3/S5 should be shown directly on “b” panel of figures. You also should consider to emphasize up- or down- regulation of nAChR subunits expression on “c” panels by highlighting subunits font as green or red according to table S2/S4/S6 data.
> the legend “smokers” and “non-smokers” has been added on Fig1/2/3-b.
> median thickness has been increased.
> statistical significance has been added where appropriate on panels b.
> As suggest by Reviewer, we emphasized the up/down regulation with a color code on panels c and Tables S1-6.
- You should decipher the abbreviation CHRNA in results (better in the introduction)
>We added the definition of the abbreviation for the genes in lines 37-39.
- Please, provide criteria for “very high” or “high” nAChR expression on lines 75 and 76 or indicate that it is mentioned in 4.5. Methods section.
>We added the note line 78.
- Some issues about immunohistochemistry:
- a) Could you provide isotype control/secondary Ab staining of a specimen as a supplementary figure?
>We added a novel Figure S2 to show immunostaining with Isotype-matched IgG antibodies.
- b) Could you justify usage of p63 or pan-cytokeratin as a stem cell marker for co-staining of different nAChRs?
>P63 is a marker of lung epithelial basal cells and the pan-cytokeratin antibody that we used specifically stained non-differentiated epithelial cells. Since we aimed to mainly provide an atlas of the subunits in the airways, we provided the co-stainings of the nAChRs with the basal cells because the main differentiated cells (ciliated and mucous-secreting) are also identifiable.
- c) Ab specificity analysis is impressive but table S8 data leads to the question: how could specific stain be guarantied in pairs: α3-nAChR and α6-nAChR; α9-nAChR and α10-nAChR; β2-nAChR and β4-nAChR?
>Reviewer is right and the absolute specificity cannot be guaranteed. This is particularly the case for subunits a2, a4, a6, a10, b2 and b4. Extensive in vitro validation should be pursued but there again the timely expression and assembly of receptors at the cell membrane would be critical.
- d) Please, provide scales for all ICH pics (including S2, if possible).
>It has been done.
- Methods section:
- a) Please, provide G instead of rpm in 4.2. section.
>It has been done
- b) For qPCR primers provide references (if possible) and amplicon length.
>Amplicon length have been added in Table S9. The primers were designed via the Universal Probe Library Assay Design Center (Roche). We added this comment lines 250-251.
- c) Provide dilutions for p63 and cytokeratin Abs. Add cat# and dilutions for secondary Abs
>It has been done.